# Developing a Yield Table and Analyzing the Economic Feasibility for *Acacia* Hybrid Plantations in Achieving Carbon Neutrality in Southern Vietnam

**Sang-Hyun Lee [1], Dong-Hyuk Kim [2], Jin-Heon Jeong [3], Seung-Hyun Han [4], Seongjun Kim [5], Hee-Jung Park [1] and Hyun-Jun Kim [6,*]**

[1]  Department of Forest Environmental Science, Chonbuk National University, Jeonju 54896, Korea
[2]  Sanrimjohap Vina (SJVINA), Hochi Minh 70000, Vietnam
[3]  Forest Geographic Information Division, Korea Forestry Promotion Institute, Seoul 07569, Korea
[4]  Forest Technology and Management Research Center, National Institute of Forest Science, Pocheon 11186, Korea
[5]  Center for Endangered Species, National Institute of Ecology, Yeongyang 36531, Korea
[6]  Department of Forest Resources, Chonnam National University, Gwangju 61187, Korea
*  Correspondence: hjkim0837@jnu.ac.kr; Tel.: +82-62-530-2082

**Abstract:** To achieve the goal of carbon neutrality, overseas plantation projects have been conducted in several countries, including Vietnam. In the present study, stand growth models and a yield table were developed and used to analyze the economic feasibility of *Acacia* hybrid plantations in southern Vietnam. Diameter at breast height, tree height, and number of trees were determined in the field; basal area, mean height, dominant tree height, stand density, and stand volume were estimated using in situ data. The initial number of trees increased for five years and reached 4947 trees ha$^{-1}$; tree numbers then decreased to 1987 trees ha$^{-1}$ until the stands reached ten years of age. The highest current annual increment of stand volume was shown to be 46.23 m$^3$ ha$^{-1}$ yr$^{-1}$ when stands were 7 years old. For 7 years of stand age, the net present value was USD 1566/ha, and the internal rate of return was 13%, exceeding the market interest rate (4%). Therefore, an *A.* hybrid plantation is a reasonable proposition for an overseas plantation project in southern Vietnam and the growth models will be useful for the management of an *A.* hybrid plantation.

**Keywords:** benefit-cost ratio (B/C ratio); internal rate of return (IRR); net present value (NPC); site index; stand growth model

## 1. Introduction

In order to limit the global temperature rise to within 1.5 °C, it is necessary to transition to a carbon-neutral society to attain net zero carbon emissions by 2050. Carbon neutrality is a concept wherein greenhouse gas emissions caused by human activities are minimized, absorbed (by forests, grasslands, and oceans, etc.), and removed (by carbon capture, utilization, and storage) to reduce the overall emissions of carbon to zero.

Many countries conducted overseas plantation projects, especially America, Africa, and Asia, specifically Laos, Indonesia, and Vietnam, to meet the goal of carbon neutrality [1]. Such overseas plantation projects could provide economic benefits to the host countries. First, it is possible to provide a stable supply of wood resources through overseas plantations. This implies that tropical areas can provide timber by cultivating the growing fast species. This not only maintains the forest processing industry, but can also act as a base for effectively managing forest resources. Second, carbon credits secured by forestation can help maintain domestic industries by offsetting their high carbon emissions. As many countries have a limited area available for forestation, overseas plantations could greatly contribute to managing forest resources and securing carbon credits. In addition, in some

cases, profits can be expected from selling these carbon credits. Thus, tropical regions are regarded as promising areas for overseas plantation projects [2].

The lack of information on local investment environments inhibits participation in overseas plantation projects [3]. The economic analysis of plantation investment is the process of estimating the expected economic costs and benefits of new projects to ensure that the investment is rational and valid. In other words, evaluating the current and future asset values of plantations is the most important task when expanding overseas plantations. Accurate growth forecasting is critical, and tree growth is essential for this evaluation. South Korea recently renewed its stem volume tables for 10 major plantation species and analyzed the effectiveness of these new stem volume tables for estimating forest stocks [4,5]. Japan and Finland also reappraised their recent stocks to predict forest growth more accurately using yield tables [6]. The forest yield table is designed mainly for the application to even-aged forest. This yield table normally displays the values for stand age, top height, number of trees per hectare, mean diameter at breast height, basal area per hectare, mean volume per hectare, and mean annual increment based on standing trees rather than cumulative value; for example, the values effectively lost due to mortality are not included. The forest yield table has limited application to different forest areas because of different biotic and abiotic conditions. However, southern Vietnam does not currently have accurate yield tables for predicting the stand growth of its major plantation species, including *Acacia* hybrids [7].

Vietnam currently maintains over 230,000 ha of *Acacia* hybrid plantations (*Acacia mangium* × *Acacia auriculiformis*) within a total *Acacia* plantation area of over 400,000 ha, as reported by the Vietnam Ministry of Agriculture and Rural Development [8]. This *Acacia* hybrid species is planted in the southern provinces of Vietnam and is now a preferred species for industrial plantations [9]. Its nitrogen fixation ability can improve infertile soil, which is economically beneficial to subsequent plantations [10]. The main stem of the hybrid is straighter than that of *A. auriculiformis* and has no angles or ribs, unlike *A. mangium* [11,12]. Additionally, predicting carbon sinks and issuing carbon credits for *Acacia* hybrid plantation are important actions as they can increase the potential income for smallholder farmers [7].

Therefore, the present study aimed to (1) develop a forest yield table capable of evaluating and predicting stand growth for *Acacia* hybrid plantations in Vietnam and (2) analyze the economic feasibility of the plantation project using the basic data from the yield table.

## 2. Materials and Methods

### 2.1. Study Sites

The study sites were located in the Xuyên Mộc district, in the province of Bà Ria-Vũng Tàu, Vietnam (10°64′48″ N, 107°42′62″ E) (Figure 1). Xuyên Mộc lies roughly 91 km from Hồ Chí Minh, and the capital can be reached in 2 h by car. *Acacia* and rubber trees are the main plantation crops in the area, comprising areas of approximately 400 and 500 ha, respectively. The annual mean temperature is 26–28 °C, and annual mean precipitation is approximately 3000 mm. Forest surveys were conducted at 11 plots from 21 October to 1 November 2019. A total of 11 plots were selected based on stand age, ranging from 1 to 9. One-year-old seedlings were planted at all plantation areas, in which this study was conducted, with a stand density of 2200 trees ha$^{-1}$. Forest survey results showed soil texture of the study area to be sandy clay loam with 28% clay, 7% silt, and 65% sand. Acidity of soil was shown as pH 4.7–5.5. Mean organic matter was less than 1% and total nitrogen was 0.09%.

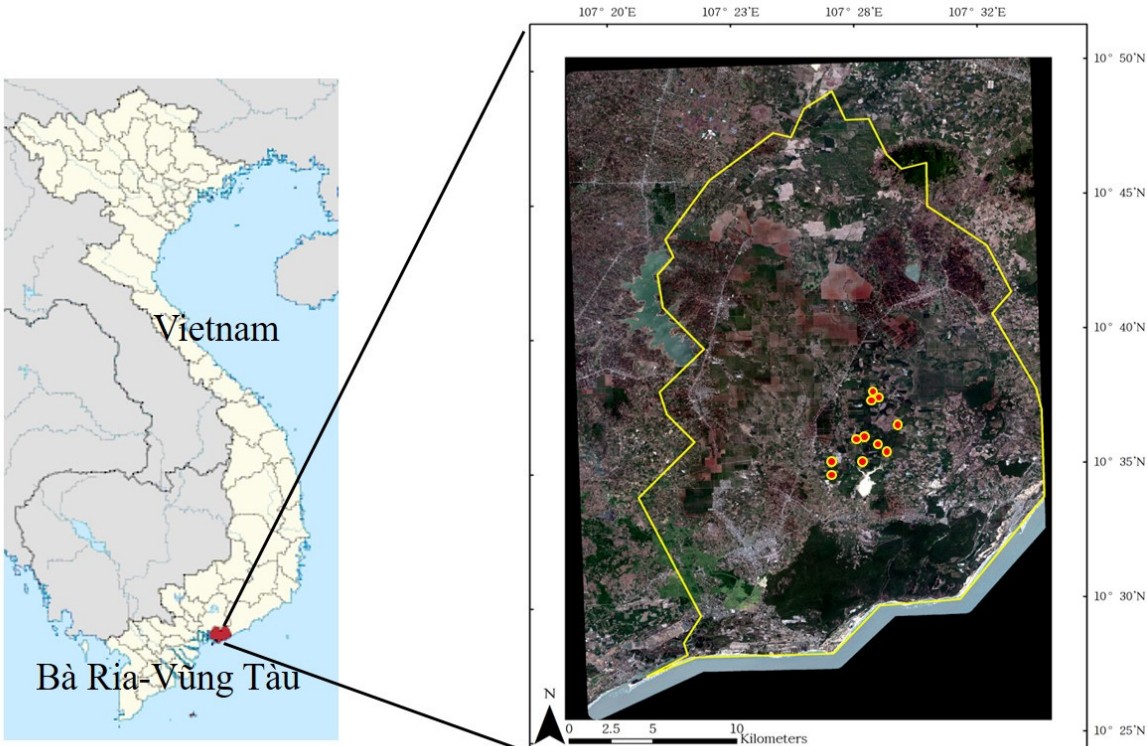

**Figure 1.** Location of plots for the study site. The red circles with yellow borders in the right-hand figure indicate the locations of the plots where forest surveys were conducted.

### 2.2. Measurement of Stand Growth

All trees within each 20 m × 20 m plot were measured to estimate stand growth. Diameter at breast height (DBH) was measured using a diameter tape (Shinill Science, South Korea), and height (H) was measured using a height meter (Suunto, Finland). Canopy class was divided into 3 classes (dominant, intermediate, suppressed), and dominant height ($H_{dom}$) was calculated as the average of the height measurements for trees in the dominant canopy class. However, for one-year-old seedlings, the root collar diameter (RCD) was measured at 2 cm height from the ground instead of the DBH. RCD was used to calculate the volume of small seedlings with a height of <1.5 m and was not used for the DBH model. The volume of each individual tree was calculated using the form factor method, as shown in Equation (1).

$$V = BA \times H \times f, \tag{1}$$

where V is the tree volume with bark ($m^3$), BA is the basal area at breast height ($m^2$), H is the tree height (m), and f is the breast-height form factor. The breast-height form factor generally varies among species and ages, ranging from 0.3 to 0.7 [13–15]. However, this study selected 0.49 as the breast-height form factor, which was developed by the Forest Science Institute of Vietnam and is used to estimate stand volume as a representative value in Vietnam [8].

### 2.3. Site Index Classification Curves

The SAS ver. 9.3 statistical program was used to estimate site indices using $H_{dom}$. The main standard statistical method consisted of the nonlinear least squares regression method via the PROC NLIN procedure. The applied dominant tree function equation is an asymptotic value, which represents the maximum value that the dependent tree can reach and produces a sigmoid curve with two inflection points, representing typical growth

patterns [16]. Through regression analysis of the heights of the dominant trees examined by age group, the $H_{dom}$ for each age can be estimated using Equation (2):

$$\text{Chapman-Richards equation: } H_{dom} = \alpha \times (1 - \exp(-\beta \times age))^{\gamma}, \tag{2}$$

Here, $H_{dom}$ is the dominant height (m), and age is the stand age (years). $\alpha$, $\beta$, and $\gamma$ are parameters. Substituting the standard age (Asi) for site index (SI) instead of stand age in the $H_{dom}$ equation, the site index equation can be derived, as in Equation (3):

$$SI = \alpha \times (1 - \exp(-\beta \times Asi))^{\gamma} \tag{3}$$

In addition, combining Equations (2) and (3) in the form of an algebraic differential equation derives the SI classification curve, which can produce various $H_{dom}$ curves according to SI, Asi, and stand age.

$$H_{dom} = SI \times (1 - \exp(-\beta \times age))^{\gamma} / (1 - \exp(-\beta \times Asi))^{\gamma} \tag{4}$$

*2.4. Developing Stand Growth Models*

Through regression analysis, the relationships among age, DBH, basal area (BA), mean tree height ($H_m$), number of trees per hectare ($N_{ha}$), and stand volume per hectare ($V_{ha}$) were derived to predict the status of stands over time. At this time, the sigmoid curve-type Chapman-Richards Equation (5) was used to predict DBH, BA, $H_m$, and $V_{ha}$, and the peak curve-shaped Gaussian Equation (6) was used to predict $N_{ha}$. The Chapman–Richards function is a popular model for describing the cumulative growth of tree and forest stands over time, so it was used in many forest-growth modelers. The Gaussian function is the probability density function of normal distribution, and is also used as the frequency curve. Thus, these functions were selected to understand and predict the stand growth of the *Acacia* hybrid plantation area.

$$F(x) = \alpha \times (1 - \exp(-\beta \times x))^{\gamma} \tag{5}$$

$$F(x) = \alpha \times \exp(-0.5 \times ((x - \beta)/\gamma)^2) \tag{6}$$

*2.5. Analyzing the Economic Feasibility*

The analysis scenario was set as follows: (1) *Acacia* hybrids were planted at a density of 2200 seedlings ha$^{-1}$; (2) the yield table developed in this study was used to calculate the growth of stand volume; (3) the sales type was pulpwood (rate of wood conversion was 90%, market price USD 30 m$^{-3}$); (4) the latest Vietnam interest rate was applied, which was 4%; and (5) logging and sales were carried out in the 7th year after plantation. The annual costs of tillage, materials, machines, labor, transport, supervision, planning, management, and land leases were also calculated for the investment analysis (Table 1). The data used in the analysis scenario were collected from interviews with a management company and a market roundtable. Net present value (NPV), benefit/cost ratio (B/C ratio), and internal rate of return (IRR) were used to analyze the economic feasibility of the *Acacia* hybrid plantation in southern Vietnam.

$$NPV = \sum_{t=0}^{n} \frac{B_t}{(1+r)^t} - \sum_{t=0}^{n} \frac{C_t}{(1+r)^t} \tag{7}$$

$$B/C \ ratio = \sum_{t=0}^{n} \frac{B_t}{(1+r)^t} / \sum_{t=0}^{n} \frac{C_t}{(1+r)^t} \tag{8}$$

$$IRR = \sum_{t=0}^{n} \frac{B_n}{(1+r)^n} = \sum_{t=o}^{n} \frac{C_n}{(1+r)^n} \tag{9}$$

**Table 1.** Annual costs per ha of production and management parts (unit: USD ha$^{-1}$).

| Years after Plantation (Years) | | 0 | 1 | 2 | 3 | 4 | 5 | 6 |
|---|---|---|---|---|---|---|---|---|
| I. Production cost | | | | | | | | |
| | Tillage | 564 | | | | | | |
| | Seedlings | 156 | 14 | | | | | |
| | Machine | 61 | 30 | 24 | 24 | 24 | 24 | |
| | Labor | 293 | 240 | 24 | 24 | 24 | 24 | 24 |
| | Transport | 43 | | | | | | |
| | Subtotal | 1117 | 284 | 48 | 48 | 48 | 48 | 24 |
| II. Management cost | | | | | | | | |
| | Supervision | 34 | | | | | | |
| | Plan | 9 | | | | | | |
| | Management | 112 | 28 | 5 | 5 | 5 | 5 | 2 |
| | Land lease | 148 | 148 | 148 | 148 | 148 | 148 | 148 |
| | Subtotal | 303 | 176 | 153 | 153 | 153 | 153 | 150 |
| Total | | 1420 | 460 | 201 | 201 | 201 | 201 | 174 |
| Cumulative total | | 1420 | 1880 | 2081 | 2282 | 2483 | 2683 | 2858 |

Here, t is time (year), n is the total period (years), B is income (USD per hectare), C is investment (USD per hectare), and r is the latest interest rate (%).

## 3. Results and Discussion

### 3.1. Analysis of Site Index

A stand age of 20 or 30 years is generally defined as the standard age for SI, but this is not fixed [17]. In the present study, a stand age of 10 years was used as the SI standard because harvesting occurs 8 years after planting in Vietnam. The results of the regression analysis of the $H_{dom}$ growth model using the Chapman–Richards equation are shown in Figure 2.

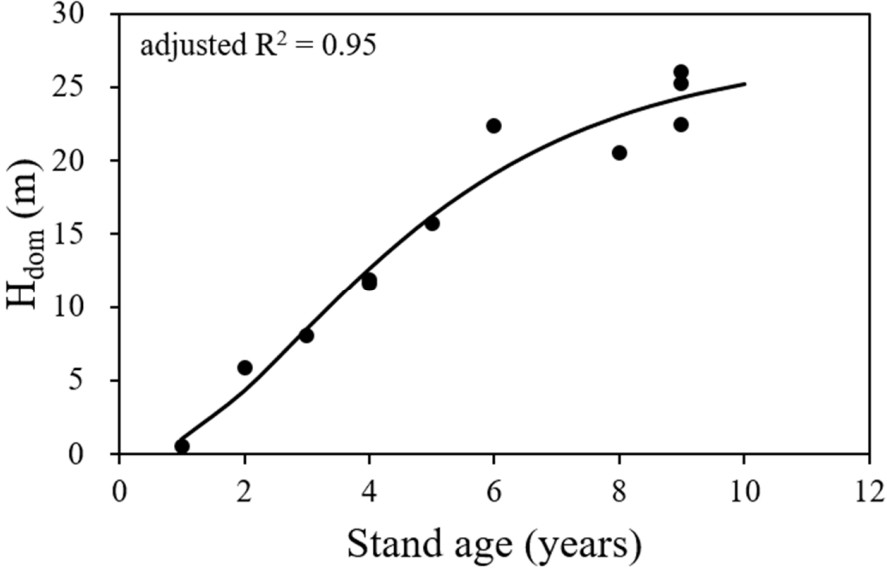

**Figure 2.** $H_{dom}$ growth model using the Chapman–Richards equation.

The average SI was analyzed using the $H_{dom}$ growth model based on age, and the SI of the study area was found to be 25.2. Therefore, the SI of the plantation area in Xuyên Mộc, in Bà Ria-Vũng Tàu, Vietnam, was determined to be 25, and the SI classification curves were derived by classifying the SI as 21, 23, 25, 27, or 29 (Figure 3). In terms of dominant height, its mean annual increment increased steadily for the first 3 years, and then decreased. The highest mean annual increment of *Acacia* hybrid was 4.2 m per year, at 3 years of age after planting. The lowest mean annual increment was 0.9 m per year at 10 years. This vigorous growth in the early growing stages and reducing growth with stand age in *Acacia* species were reported from previous studies [18,19].

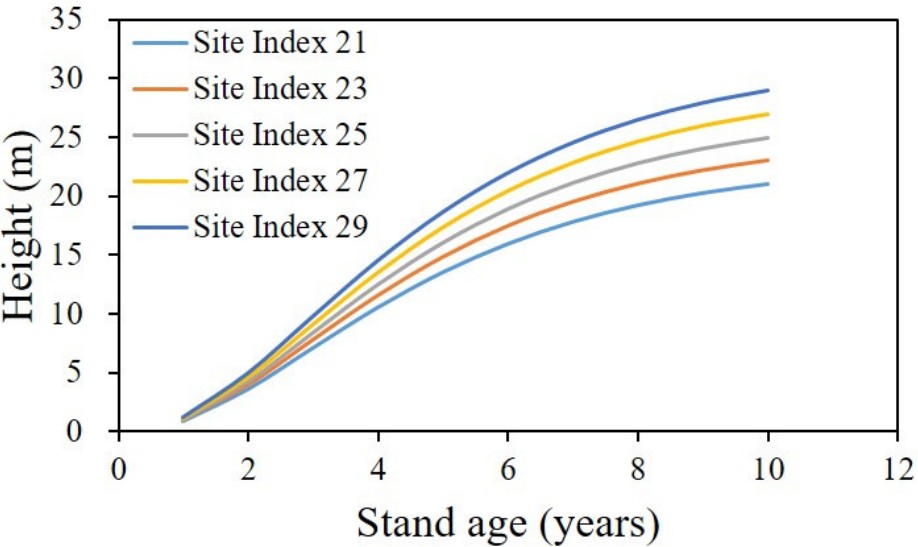

**Figure 3.** Site index classification curves for *Acacia* hybrid plantations in Vietnam.

### 3.2. Developing Stand Growth Models

Forest stands are complex environments composed of similar individuals. Predicting stand-scaled growth differs from predicting single tree growth because it is an important parameter for setting forest management goals. In order to predict the stand growth for these forests, DBH, BA, $H_m$, $N_{ha}$, and $V_{ha}$ were measured in Vietnamese *Acacia* hybrid plantations (Table 2). In general, in Xuyên Mộc, 1-year-old seedlings are planted and then harvested at 9 years of age. Thus, the stand growth rate was predicted until the stand age of 10 years (Figure 4). As a result, DBH, $H_m$, $H_{dom}$, and $V_{ha}$ showed remarkable increases as stand age increased. However, $N_{ha}$ peaked at 5275 in plot 5, which was planted in 2016, and similar values were found at plots 3, 4, and 7, although their stand ages were different. Various growth models were developed using these data.

The parameters for each growth model determined for *Acacia* hybrid plantations in Vietnam were estimated (Table 3). Even though the $N_{ha}$ model had a low fitness (adjusted $R^2$ = 0.88; $p < 0.0001$), other models had acceptable goodness of fit (adjusted $R^2$) at 0.90 or more, indicating the validity of these models for evaluating and predicting stand growth ($p < 0.0001$).

The DBH of the *Acacia* hybrid increased with stand age (Figure 4a). In particular, trees were expected to grow up to 12.5 cm in DBH by 9 years after planting (10 years old). In normal forests, DBH increases rapidly in early stages but gradually slows down after maturation. The DBH growth rate gradually decreased over time after trees were 6 years old. Generally, pruning can increase the DBH growth rate of the remaining stems [20]. The incidence of large trunks and the mean growth of individual trees are affected by the initial stand density [21], so pruning can lead to increased DBH growth. In addition, thinning provides an opportunity to maintain a high growth rate of the remaining trees by removing poorly formed trees. Thus, it is necessary to improve the economic efficiency of plantations by analyzing growth differences under different stand densities and pruning practices.

**Table 2.** Summary of age, diameter at breast height (DBH), mean height ($H_m$), dominant height ($H_{dom}$), number of trees per hectare ($N_{ha}$), and stand volume per hectare ($V_{ha}$) for each plot.

| Plots | Stand Age (Years) | DBH (cm) | $H_m$ (m) | $H_{dom}$ (m) | $N_{ha}$ (Trees ha$^{-1}$) | $V_{ha}$ (m$^3$ ha$^{-1}$) |
|---|---|---|---|---|---|---|
| 1 | 1 | - | 0.5 | 0.5 | 2200 | 1 |
| 2 | 2 | 4.3 | 5.2 | 5.9 | 4000 | 16 |
| 3 | 3 | 5.4 | 7.5 | 8.0 | 5025 | 47 |
| 4 | 4 | 6.6 | 10.6 | 11.6 | 5000 | 95 |
| 5 | 4 | 6.2 | 10.4 | 11.9 | 5275 | 86 |
| 6 | 5 | 8.8 | 11.7 | 15.7 | 3175 | 124 |
| 7 | 6 | 7.8 | 12.6 | 22.4 | 5075 | 177 |
| 8 | 8 | 9.3 | 15.5 | 20.5 | 4525 | 257 |
| 9 | 9 | 11.8 | 18.8 | 26.0 | 2350 | 276 |
| 10 | 9 | 12.8 | 16.7 | 22.4 | 2450 | 300 |
| 11 | 9 | 12.3 | 19.1 | 25.3 | 2800 | 357 |

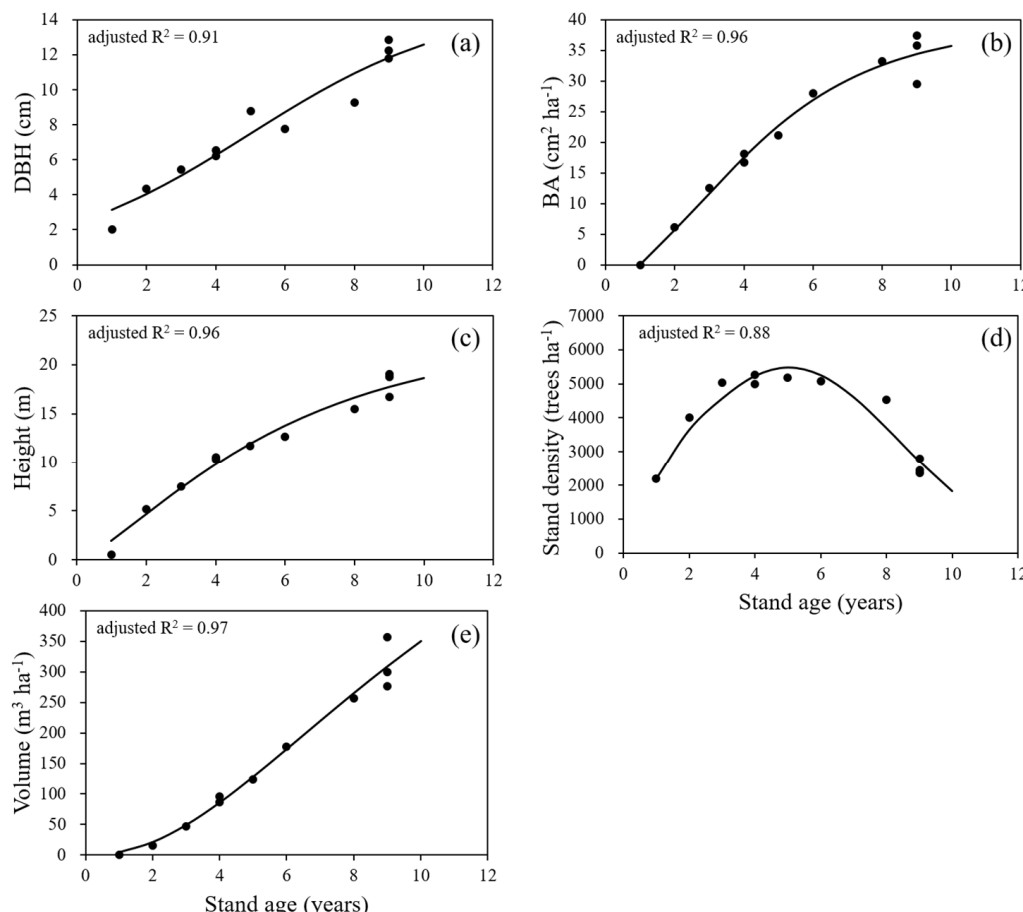

**Figure 4.** Stand growth curves: (**a**) diameter at breast height (DBH); (**b**) basal area (BA); (**c**) mean height ($H_m$); (**d**) stand density; and (**e**) stand volume, over stand age for *Acacia* hybrid plantations in southern Vietnam.

**Table 3.** Parameters and R$^2$ of stand growth equations for an *Acacia* hybrid plantation in Vietnam.

| Model | Parameter | | | Adjusted R$^2$ | $p$ | MSE |
|---|---|---|---|---|---|---|
| | $\alpha$ | $\beta$ | $\gamma$ | | | |
| DBH | 17.8909 | 0.1323 | 1.1519 | 0.91 | <0.0001 | 1.3 |
| BA | 39.0736 | 0.3451 | 2.7718 | 0.96 | <0.0001 | 5.3 |
| H$_m$ | 22.7448 | 0.2067 | 1.4696 | 0.96 | <0.0001 | 1.2 |
| H$_{dom}$ | 27.5182 | 0.3398 | 2.6249 | 0.95 | <0.0001 | 3.7 |
| N$_{ha}$ | 5479.0401 | 5.0264 | 3.3612 | 0.88 | <0.0001 | 203,231.2 |
| V$_{ha}$ | 705.0696 | 0.1414 | 2.5086 | 0.97 | <0.0001 | 460.5 |

The basal area at breast height affects stand volume and is generally used to estimate the degree of stocking. The degree of stocking is a measure of the ratio of in situ BA to BA in the yield table. Basal area at breast height is a major parameter used to estimate stand volume, stand growth, and degree of stocking. In addition, the DBH model strongly affects changes in average values, not the actual growth conditions of the stand, whereas the BA model provides information about real changes in stand conditions. For this reason, BA can be considered a more important factor than DBH when evaluating stand growth.

As a result, the BA growth curve showed a sigmoid morphology, which is a typical growth pattern (Figure 4b). Up to 5 years after planting, the growth rate increased gradually over time because of the increase in both DBH and N$_{ha}$. After this time, the annual growth rate of BA decreased owing to reductions in the annual growth of DBH and N$_{ha}$. However, despite the decrease in the annual increment of BA, the annual increment of V$_{ha}$ increased continuously, suggesting that the effect of stand density on annual V$_{ha}$ growth rate was not significant after 5 years of age.

Height is an important variable when evaluating stand growth in even-aged forests, as it is relatively independent of stand density and is much affected by thinning. For this reason, mean height is used to measure the productivity of forests and to assess the development of the stand structure, while dominant height is used to evaluate the production capacity and site quality. The H$_m$ of the *Acacia* hybrid in Vietnam continued to increase with increasing stand age, reaching a maximum of 18.6 m at 10 years of age (Figure 4c). Forest management seems to be required to improve the stand growth of *Acacia* hybrid considering the dominant height of 25.2 m at 10 years. In contrast, the production system in southern Vietnam does not include any management system, such as improvement cutting and thinning. This nonintervention system can prevent the erosion of the topsoil and dryness of the land, as well as reduce the decline in soil productivity caused by harvesting. Despite the advantages of the nonintervention system, improvement cutting and thinning can enhance the mean annual growth of DBH and mean height through the decrease in competition for space, nutrients, and photosynthesis among the individuals. Thus, further study should be conducted to analyze the effects of stand density on stand growth, especially the stand volume per hectare.

Hai et al. (2008) found that environmental conditions affect the growth of *Acacia* species more than genetic factors [22]. *Acacia* hybrids were shown to grow faster in Vietnam, especially in low-latitude provinces [23]. Kha et al. (2005) and Kha (2009) also confirmed the potential for wood production from *Acacia* hybrids in southern Vietnam [9,24]. Thus, the environmental conditions in Xuyên Mộc may improve the growth of *Acacia* hybrids.

Furthermore, Dung et al. (2005) reported that pruned *Acacia* hybrid trees were significantly taller than non-pruned trees at 3 years after pruning treatment [25]. CARD (2005) recommended that the first pruning of *Acacia* hybrid plantations should be done when the canopy closes in Vietnam [26].

Among the factors affecting stand growth, the number of remaining trees has a direct influence on DBH, BA, and V$_{ha}$. For this reason, N$_{ha}$ is an important factor to consider when estimating stand density and growth. Even though 2200 seedlings per hectare were

initially planted, $N_{ha}$ continuously increased in the first 4 years after plantation, and finally reached approximately 5500 trees per hectare at the stand age of 5 years (Figure 4d). This situation may result from the newly formed sprouts. Sprouts were observed once the stand age was 2 years. This sprouting is an inherent characteristic of *Acacia* species [27]. In stands older than 5 years, the number of remaining trees sharply decreased and reached 1987 trees ha$^{-1}$ at a stand age of 10 years. In other words, the number of remaining trees increased by approximately 225% compared with the number of seedlings planted during the first 5 years but decreased by approximately 249% in the last 5 years. Sprouts have a very rapid initial growth rate compared with the growth rate of seedling stem. As a result, the stem, which was 1 at the time of planting, increased to 3 to 5 by the stand age of 5 years. When the stand age is older than 5 years, each individual tree strongly competes for growth space due to the rapidly increasing stand density. The high competition rate leads to the high mortality rate. Moreover, the increase in the number of sprouts may restrain the diameter growth rate, including that of both seedlings and sprouts. These results indicate that maintaining an appropriate stand density through improvement cutting promotes stand growth in *Acacia* hybrid plantation areas. Therefore, further research is needed to examine the effects of thinning on stand growth.

However, *Acacia* hybrid individuals may differ from one another in terms of form and quality, resulting in a gradually decreasing $N_{ha}$ due to natural death. Therefore, in order to generate high income, it is necessary to standardize timber trees by developing and producing superior varieties. This standardization of forest production is expected to improve forest productivity in Vietnam and consequently increase the economic efficiency of plantations.

The annual growth and average growth of stand volume are very important when evaluating forest growth because they can be used as measures of production capacity. In particular, stand volume indicates the accumulated volume, which can predict the total growth and economic value of stands. The stand volume is generally represented per hectare and can vary depending on the stand type, species, mixing efficiency, and age. However, the inherent site characteristics of soil and climate are more important than fertilizer application in determining stand growth rates [28]. For example, the relation between the growth development of *Acacia mangium* and clay content was negative [29], and the fast growth of the *Acacia* hybrid is commonly observed in decreasing latitudes [28].

In the *Acacia* hybrid plantation in Vietnam, BA and $H_m$ tended to increase with increasing stand age, and $N_{ha}$ sharply decreased due to strong competition for growth space after stands reached 5 years of age. Nevertheless, $V_{ha}$ steadily increased until the stand age was 9 years (Figure 4e). This may be due to the high growth rate of DBH and $H_m$ caused by the tropical climate. At a stand age of 10 years, $V_{ha}$ increased up to 351 m$^3$ ha$^{-1}$ and continued to increase steadily. Thus, further studies are required to analyze the economic feasibility of various felling seasons.

### 3.3. Developing a Stand Yield Table for Acacia Hybrid Plantations in Vietnam

Acquiring information on stand growth, as well as stand volume, is essential, and the development of high accuracy and species-specific yield tables is critical for forest management. In the United States, a species-specific yield table for each major species exists for each state, and this is used to plan various forest management activities and studies. Currently, various species are planted and harvested in Vietnam, and it is, therefore, necessary to develop yield tables in order to provide information on plantation areas. Accordingly, the present study prepared a "stand yield table (site quality—middle)" to provide professional information regarding *Acacia* hybrid species in the current plantation areas in Vietnam (Table 4).

**Table 4.** Yield table developed in the present study for *Acacia* hybrid plantations in Vietnam. Diameter at breast height (DBH), basal area (BA), mean height ($H_m$), dominant height ($H_{dom}$), number of trees per hectare ($N_{ha}$), volume per hectare ($V_{ha}$), and current annual increment (CAI) were calculated against stand age using the growth models developed in this study.

| Stand Age (Years) | DBH (cm) | BA (m² ha⁻¹) | $H_m$ (m) | $H_{dom}$ (m) | $N_{ha}$ (Trees ha⁻¹) | $V_{ha}$ (m³ ha⁻¹) | CAI (m³ ha⁻¹ yr⁻¹) |
|---|---|---|---|---|---|---|---|
| 1 | 1.6 | 1.4 | 0.5 | 0.5 | 2877 | 4 | 4 |
| 2 | 3.3 | 5.9 | 3.9 | 4.3 | 3663 | 21 | 17 |
| 3 | 4.9 | 11.7 | 7.3 | 8.5 | 4345 | 49 | 28 |
| 4 | 6.4 | 17.5 | 9.8 | 12.6 | 4803 | 86 | 37 |
| 5 | 7.8 | 22.7 | 11.9 | 16.2 | 4946 | 128 | 42 |
| 6 | 8.9 | 26.8 | 13.8 | 19.1 | 4747 | 174 | 46 |
| 7 | 10.0 | 30.1 | 15.3 | 21.3 | 4245 | 220 | 46 |
| 8 | 11.0 | 32.6 | 16.6 | 23.0 | 3537 | 265 | 45 |
| 9 | 11.8 | 34.5 | 17.7 | 24.3 | 2746 | 309 | 44 |
| 10 | 12.5 | 35.8 | 18.6 | 25.2 | 1986 | 351 | 42 |

A stand yield table is used to estimate the growth and yield of stands and provides information on DBH, BA, $H_m$, $H_{dom}$, $N_{ha}$, $V_{ha}$, and current annual increment (CAI). The highest CAI for *Acacia* hybrids in Vietnam was shown to be 46.23 m³ ha⁻¹ yr⁻¹ at 7 years of stand age. After this age, CAI decreased slightly because of decreases in the annual growth of DBH, $H_m$, and $N_{ha}$. In particular, CAI decreased after stands reached 8 years of age owing to decreased $N_{ha}$. From the previous study [29], the CAI for *A. mangium* also significantly decreased due to a low survival rate and stand density since stand age was 8.5 years. Thus, it is desirable that harvesting should be carried out at a stand age of 7 years, as the increase in economic value with time was shown to decrease after a stand age of 8 years. The productivity of *Acacia* hybrids at 5 years of age exceeded 30 and 15 m³ ha⁻¹ yr⁻¹ in the Binh Duong and Ba Vi regions of Vietnam, respectively [24,30]. Therefore, the environmental conditions in Vietnam may be suitable for future *Acacia* hybrid plantations, particularly in Bà Ria-Vũng Tàu.

Conversely, Son et al. (2010) compared the stand growth of *Acacia* hybrid plantations in Vietnam to that of *A. mangium* plantations in Kalimantan Tengah, Indonesia, and found that the average DBH in Vietnam (12.5 cm) was significantly lower than that in Indonesia (24.4 cm) at 10 years of stand age [1]. However, a total of 2200 trees ha⁻¹ were planted initially in the Vietnamese plantation, whereas only 1000 trees ha⁻¹ were planted in Indonesia. This difference in $N_{ha}$ between Vietnam and Indonesia may have caused the observed difference in DBH. Mean height was also higher in Indonesia (24.7 m) than that in Vietnam (18.6 m). However, the growth of $V_{ha}$ in southern Vietnam (351 m³ ha⁻¹) was greater than in Indonesia (243 m³ ha⁻¹) because of the higher stand density. In addition, the average CAIs were shown to be 35.05 m³ ha⁻¹ yr⁻¹ in southern Vietnam and 24.31 m³ ha⁻¹ yr⁻¹ in Indonesia for 10 years after planting. The highest CAIs in southern Vietnam and Indonesia were found to be 46 m³ ha⁻¹ yr⁻¹ at 7 years and 39 m³ ha⁻¹ yr⁻¹ at 4 years, respectively. These results confirm that the rotation period should be longer in southern Vietnam than that in Indonesia.

### 3.4. Economic Feasibility of Acacia Hybrid Plantations in Vietnam

For the *Acacia* hybrid plantation in Vietnam, the range of harvest age is from about 5 years (for pulp product) to 12 years (for low productive sites) [28]. The common harvest age for *Acacia* hybrid in southern Vietnam is about 9 years. The rotation period for commercial plantations in Vietnam is based on the production system and the region. In this study, the rotation period for *Acacia* hybrid plantations in southern Vietnam was set at 7 years depending on the stand growth patterns. The stand volume of *Acacia* hybrids was found to increase to 220 m³ ha⁻¹ in 7 years, and the generated revenue was USD 5934 ha⁻¹

when all of it was sold as pulpwood (Table 5). The total investment cost of plantation for 7 years was USD 2858 ha$^{-1}$, and the net profit after 6 years of plantation was USD 3076 ha$^{-1}$ in present value, resulting in an annual income of USD 439 ha$^{-1}$. The annual income in Vietnam is lower than that in Paraguay, at USD 569 ha$^{-1}$ [31]. The net present value (NPV) was USD 2431 ha$^{-1}$. The stands turned profitable after 3 years of plantation. Vietnam has a relatively low annual investment cost due to cheap labor and land. Hence, the transition period of attaining profitability is lower in Vietnam than that in Malaysia (9 years) and Indonesia (11 years) [32,33]. Additionally, the capital turnover in Vietnam (7 years) is faster than that in Paraguay (12 years) [31]. The internal rate of return (IRR) for 7 years in Vietnam was 13%, exceeding the market interest rate (4%). Thus, it is concluded that overseas plantation projects in Vietnam are economically feasible.

**Table 5.** Summary of annual income, investment, and cash flow and the results of economic feasibility analysis for *Acacia* hybrid plantations in southern Vietnam. NPV is net present value, B/C Ratio is benefit-cost ratio, and IRR means internal rate of return.

| Stand Age (Years) | Income (USD ha$^{-1}$) | Investment (USD ha$^{-1}$) | Cash Flow (USD ha$^{-1}$) | NPV | B/C Ratio | IRR (%) |
|---|---|---|---|---|---|---|
| 1 | 0 | 1420 | −1420 | −1420 | 0.00 | |
| 2 | 566 | 1880 | −1314 | −1263 | 0.30 | |
| 3 | 1325 | 2081 | −756 | −699 | 0.64 | |
| 4 | 2318 | 2282 | 36 | 32 | 1.02 | |
| 5 | 3462 | 2483 | 979 | 837 | 1.39 | |
| 6 | 4686 | 2683 | 2003 | 1646 | 1.75 | |
| 7 | 5934 | 2858 | 3076 | 2431 | 2.08 | 13 |

## 4. Conclusions

This study was conducted to develop stand growth models and a yield table for *Acacia* hybrid plantations in Vietnam in order to provide the basic information needed to analyze the economic feasibility of overseas plantation projects. First, stand growth was characterized by developing stand growth models. Results show that the growth pattern of the *Acacia* hybrid consists of two steps: fast growing in the first step and a gradual decrease in growth in the second step. This trend is consistent with the results from the provenance trials in several tropical countries. Secondly, when comparing the results from this study with others, the trend of stand growth of the *Acacia* hybrid is often an acceleration as latitude decreases.

The stand volume reached over 350 m$^3$ ha$^{-1}$ for plantations in southern Vietnam at a stand age of 10 years, which was higher than that found for plantations in northern Vietnam and Indonesia. The stand growth, in terms of DBH and mean height, in southern Vietnam was more vigorous than that in northern Vietnam, but less than that in Indonesia. However, the stand volume in southern Vietnam was higher than that in Indonesia due to its higher stand density. These results indicate that the inherent environmental conditions are an important factor in the growth trend of *Acacia* hybrid plantations, and controlling the stand density can enhance the value of a plantation. Thus, further studies should be conducted to analyze the effects of thinning on stand growth for maximizing the economic value of *Acacia* hybrid plantations.

Furthermore, the plantation project in southern Vietnam showed high feasibility and had several advantages. First, *Acacia* hybrids grow faster in tropical countries, such as Vietnam, than in the others. Second, the turnover of funds is rapid as the final age is less than 10 years. Third, labor costs are low as there is abundant idle labor. Finally, land rent is also reasonable. For these reasons, other species should be studied in further research.

**Author Contributions:** Conceptualization, S.-H.L., D.-H.K. and H.-J.K.; methodology and investigation, D.-H.K. and J.-H.J.; writing—original draft preparation, S.-H.H., S.K. and H.-J.P.; writing—review

and editing, H.-J.K.; supervision, H.-J.K.; project administration, H.-J.K.; funding acquisition, H.-J.K. All authors have read and agreed to the published version of the manuscript.

**Funding:** This research was funded by Chonnam National University (grant number: 2021–2119) and the Korea Forest Service (grant number: 2020183C10-2222-AA02).

**Institutional Review Board Statement:** Not applicable.

**Informed Consent Statement:** Not applicable.

**Data Availability Statement:** Not applicable.

**Conflicts of Interest:** The authors declare no conflict of interest.

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
