# Peer review of "Developing a Yield Table and Analyzing the Economic Feasibility for Acacia Hybrid Plantations in Achieving Carbon Neutrality in Southern Vietnam"

_forests, doi:10.3390/f13081316_

Round 1

Reviewer 1 Report

The manuscript requires another round of revision after some improvements suggested in the attached document. The paper has merit in filling an information gap for the management of A. hybrid plantations in southern Vietnam, but I believe the authors need to address some important methodological and analytical issues before the research can be considered for publication. 

Thank you,

Reviewer 2 Report

Dear Publishers and Authors,

I suggest

a) revide all the economic analyses of the manuscript;

b) support this section with appropriate scientific literature;

c) verify if the Acacia hibryd plantation cover also the income loss from the activities developed before the Acacia hibryd plantation has been planted

c) give greater transparency to economic processing in terms of data and formulae;

Other comment are included in the file attached  

Round 2

Reviewer 1 Report

Thank you to the authors for sending the revised version and their effort and edits to improve the manuscript. I agree with most of the answers to my comments; however, a few minor comments remain: 

Thank you,

Reviewer 2 Report

I have done before (see the comment for Editors)

Author Response

We appreciate the reviewer's comments. Thank you very much for your time and efforts.